# Corrosion Resistance of Heat-Treated Ni-W Alloy Coatings

**DOI:** 10.3390/ma13051172

**Published:** 2020-03-06

**Authors:** Magdalena Popczyk, Julian Kubisztal, Andrzej Szymon Swinarew, Zbigniew Waśkiewicz, Arkadiusz Stanula, Beat Knechtle

**Affiliations:** 1Faculty of Science and Technology, University of Silesia in Katowice, 41-500 Chorzów, Poland; magdalena.popczyk@us.edu.pl (M.P.); julian.kubisztal@us.edu.pl (J.K.); andrzej.swinarew@us.edu.pl (A.S.S.); 2Institute of Sport Science, The Jerzy Kukuczka Academy of Physical Education, 40-065 Katowice, Poland; z.waskiewicz@awf.katowice.pl (Z.W.); a.stanula@awf.katowice.pl (A.S.); 3Department of Sports Medicine and Medical Rehabilitation, Sechenov University, 119991 Moscow, Russia; 4Institute of Primary Care, University of Zurich, 8091 Zurich, Switzerland

**Keywords:** Ni-W alloy coating, heat treatment, corrosion resistance

## Abstract

The paper presents research on evaluation of corrosion resistance of Ni-W alloy coatings subjected to heat treatment. The corrosion resistance was tested in 5% NaCl solution by the use of potentiodynamic polarization technique and electrochemical impedance spectroscopy. Characteristics of the Ni-W coatings after heat treatment were carried out using scanning electron microscopy, scanning Kelvin probe technique and X-ray diffraction. Suggested reasons for the improvement of properties of the heat treated Ni-W coating, obtained at the lowest current density value (125 mA∙cm^−2^), are the highest tungsten content (c.a. 25 at.%) as well as the smallest and the most homogeneous electrochemically active surface area.

## 1. Introduction

The electroplating technique is increasingly used to obtain new materials with specific functional properties. This is due to the fact that by controlling the deposition parameters, i.e. voltage, current, bath composition, temperature, it is possible to influence the structure of the obtained material, and hence its properties. The advantage of this method is the possibility of simultaneously co-depositing several metals as well as incorporation powders of metals, non-metals or chemical compounds into the coating [1,2,3,4,5,6,7,8,9,10,11,12,13,14,15,16,17,18,19,20,21,22,23,24,25,26,27,28,29,30,31,32,33,34,35,36,37,38,39,40,41,42,43,44]. Thus, the electroplating technique allows obtaining alloy and composite coatings (amorphous or crystalline) with a specific chemical and phase composition, as well as modelled surface morphology. Many metals are currently used as electrode materials in various electrochemical processes. Among them are the metals from the group of irons, especially nickel, which is characterized by good corrosion resistance and high catalytic activity in the process of hydrogen evolution. In order to improve the utilization of nickel coatings, various methods of their modifications could be applied, such as the use of alloys instead of pure elements. The interest in electrodeposited nickel - tungsten alloys is due to their specific magnetic, electrical, mechanical, thermal and corrosion properties [19,20,21,22,23,24,25,26,27,28,29,30,31,32,33,34,35,36,37,38,39,44]. These alloy coatings are widely used in the elements of machines operating under high mechanical load, at high temperatures, as well as in aggressive environments. Ni-W coatings are also used as electrode materials for hydrogen evolution reaction (HER) [2,19,40]. It should be noted that nickel - tungsten alloys can only be obtained from aqueous solutions through an induced code position, that is, tungsten is code posited with nickel. Sulphate, sulfamine and citrate baths with the addition of sodium tungstate are usually used [19,20,21,22,23,24,25,26,27,28,29,30,31,32,33,34,35,36,37,38,39,44]. 

Generally, heat treatment of electrolytic coatings should increase their corrosion resistance what was confirmed in earlier studies e.q. in [10,13,15,17,41]. The formation of new phases and the reduction of the active surface after heat treatment are the main reasons for improving the corrosion resistance of these materials. Thus, we expect that heat treatment of investigated Ni-W alloy coatings can also significantly slow down corrosion processes occurring on its surface. 

According to our knowledge there is lack of information about corrosion resistance of Ni-W coatings subjected to heat treatment in the air. Thus, the aim of this work is to study corrosion properties of heat-treated Ni-W coatings in 5% NaCl solution especially with respect to surface morphology, chemical and phase composition. The Ni-W coatings were deposited under galvanostatic conditions at the following cathodic current densities: 125, 150, 175 and 200 mA∙cm^−2^. The heat treatment of all coatings was carried out at a temperature of 1173 K. Therefore, the coatings discussed in the article were marked as follows: C125/1173, C150/1173, C175/1173 and C200/1173.

## 2. Materials and Methods

The Ni-W alloy coatings were obtained by electroplating from the electrolyte of the following composition (concentrations in g∙dm^−3^): NiSO_4_∙7H_2_O–13, Na_2_WO_4_∙2H_2_O–68, C_6_H_5_O_7_Na_3_∙2H_2_O–200 and NH_4_Cl–50. For preparation of the bath ultrapure water (Millipore, 18.2 MΩ cm) and ‘analytical grade’ reagents (Avantor Performance Materials Poland S.A.) were used. The coatings were deposited galvanostatically at the current densities 125, 150, 175 and 200 mA∙cm^−2^ and temperature of 343 K. The coatings were deposited on the steel (S235) plate of 1.0 cm^2^ geometric surface area. A platinum mesh served as an auxiliary electrode. The chemical composition of the as-deposited Ni-W alloy coatings is presented in Table 1.

Heat treatment of Ni-W alloy coatings was carried out in a muffle stove of the type FCF 2.5 SHMgO (Czylok Company, Jastrzębie-Zdrój, Poland) at 1173 K for 1 h in the air.

The surface morphology and chemical composition of the heat-treated coatings was studied using a scanning electron microscope (SEM, JEOL JSM–6480, JEOL Ltd., Tokyo, Japan) equipped with an energy dispersive spectroscopy (EDS) detector (JEOL Ltd., Tokyo, Japan). The phase composition was determined by means of X-ray diffraction method using Philips X’Pert PW 3040/60 X-ray diffractometer (U = 40 kV, I = 30 mA, Panalytical, Almelo, Netherlands) with copper radiation (λ (Cu K_α_) = 1.54178 Å). The data collection was over the 2-theta range of 20° to 120° in steps of 0.02°.

Corrosion resistance of the heat-treated coatings was determined, using potentiodynamic polarization technique and electrochemical impedance spectroscopy (EIS). These measurements were carried out in a 5 wt.% NaCl solution, using three-electrode cell and an AUTOLAB^®^ electrochemical system (PGSTAT30, Metrohm Autolab B.V., Utrecht, Netherlands). The auxiliary electrode was a platinum mesh and the reference electrode was a saturated calomel electrode (SCE). Potentiodynamic curves were recorded in the potential range ± 100 mV versus open circuit potential with rate *v* = 1 mV∙s^−1^. 

The electrochemical impedance spectroscopy was performed at the corrosion potential. In these measurements, the amplitude of the ac signal was 10 mV. A frequency range from 10 kHz to 0.1 Hz was covered with 10 points per decade. All electrochemical investigations were made at 298 K.

Contact potential difference (*CPD*) maps and surface topography maps of the heat-treated coatings were recorded by means of Scanning Kelvin Probe (SKP) technique using PAR Model 370 Scanning Electrochemical Workstation (Princeton Applied Research, Oak Ridge, USA) equipped with a tungsten Kelvin probe (KP). The scanning area was 4000 × 4000 μm^2^ and the distance between the probe and the sample was ca. 100 μm.

## 3. Results and Discussion

The heat-treated Ni-W coatings are characterized by grey, smooth and uniform surface. The surface morphology of the coatings differs, which means it depends on the deposition current density (Figure 1). The surface of C125/1173 coating shows small, separately located globules changing into larger ones with increasing of deposition current density. Coatings obtained at low current density values have a poorly developed surface. It can be explained by that low current densities favor the slow discharge of ions at electrodes, and therefore the growth rate of the resulting grains exceeds the speed of forming of new ones. As the current density increases, the rate of formation of new grains also increases what result in more developed surface. The increase in the density of the deposition current causes intense hydrogen evolution, which in turn can cause the formation of porous coatings.

The phase composition of the as-deposited Ni-W alloy coatings is independent of applied current conditions. All X-ray diffraction patterns show the presence of reflexes coming from solid solution of W in Ni. An example of X-ray diffraction pattern obtained for Ni-W coating deposited at current density of 175 mA∙cm^−2^ is shown in the Figure 2a. The phase composition of the Ni-W alloy coatings after heat treatment is also independent of applied current conditions. During the heat treatment in the air the solid solution of tungsten in nickel breaks down and chemical reaction with oxygen proceeds leading to a formation of new phases. X-ray diffraction patterns shown in Figure 2b–e indicate that the C125/1173, C150/1173, C175/1173 and C200/1173 coatings consist of three phases, i.e., Ni_4_W, WO_2_ and WO_3_.

Values of the corrosion parameters i.e. corrosion potential *E*_corr_ and corrosion current density *j*_corr_ were determined from measured dependencies *j* = *f* (*E*). It was found that the value of the corrosion potential for the C125/1173 coating is the highest compared to the *E*_corr_ obtained for the coatings deposited at larger current densities i.e. C150/1173, C175/1173 and C200/1173 (Figure 3, Table 2). It was also noted that, for the C125/1173 coating, the value of corrosion current density is lower compared to the other coatings (Table 2). This suggests that the C125/1173 coating, is more corrosion resistant in 5 wt.% NaCl solution than the other investigated coatings. It should be added that all heat-treated Ni-W coatings are characterized by the definitely higher corrosion resistance compared to the substrate (corrosion potential of S235 steel is −739 mV) [4].

The results of the EIS investigations presented in the form of Nyquist plots (−*Z*″ = *f* (*Z*′)) were shown in Figure 4. For all investigated coatings one semicircle in the whole range of frequencies is observed. It has been found that this behavior of the heat-treated Ni-W coatings could be described by one-CPE electrode model (Figure 5). This is typical model for rough or porous materials. Such equivalent circuit is characteristic for materials composed of cylindrical pores of radius *r* and length *l*. As was shown in a paper [45] for short and wide pores *l*^2^/*r* is very small and only one semicircle on the complex plane plot (Nyquist plot) was observed. The one-CPE model consists of the solution resistance *R*_s_ in series with a parallel connection of the *CPE* element (*Z*_CPE_ = 1/[(j*ω*)*^ϕ^ T*] where *T* is the capacitive parameter, *ϕ* is a dimensionless parameter and *ω* is the angular frequency of ac voltage) and the polarization resistance *R*_p_ [42]. 

Approximations of the experimental impedances using the one-CPE model allowed to determine the following parameters: *R*_p_, *R*_s_, *T*, *ϕ* (Table 3). Note that lower values of polarization resistance *R*_p_ indicate that a material is more susceptible to corrosion.

The double-layer capacitance, *C*_dl_, was calculated according to [43]:*T* = *C*_dl_*^ϕ^* (1/*R*_s_ + 1/*R*_p_)^1-^*^ϕ^*(1)

The ratio of capacitances *C*_dl_ determined for Ni-W coating and ideally smooth nickel electrode (20 μF∙cm^−2^ [43]) gives factor of electrochemically active surface area, *R*_f_ (Table 3). Larger values of this parameter indicate larger interfacial surface, and hence deterioration of material corrosion resistance. The smallest electrochemically active surface area and the highest polarization resistance obtained for C125/1173 sample clearly indicate that this coating exhibit the best anticorrosion properties compared with the other coatings.

Figure 6 shows *CPD* maps registered for the studied Ni-W coatings. Statistical analysis of the obtained maps allows determining parameters describing quantitatively the surface properties i.e. average (*CPD*_av_) and root mean square (*CPD*_q_) of contact potential difference [46,47,48]. It was stated that the C125/1173 coating (Figure 6a, Table 4) is characterized by the highest value of *CPD*_av_ which equals c.a. −1060 mV_KP_ (mV_KP_ is the voltage measured in relation to the Kelvin probe). Increasing of the deposition current density to 200 mA∙cm^–2^ (Figure 6d, Table 4) causes that the *CPD*_av_ decreases by about 140 mV_KP_. Deviation of the *CPD* values from the mean (represented by *CPD*_q_) is the smallest for C125/1173 and equals c.a. 16 mV_KP_. It means that this coating shows the most homogeneous surface of all the coatings tested. It should be noted that in the case of C200/1173 coating, obtained at the highest current density, *CPD*_q_ increases more than two times in comparison with C125/1173. Figure 7 shows the tungsten content (at.%) and corrosion current density (*j*_corr_) plotted versus the average contact potential difference (*CPD*_av_). It has been found that the increase of tungsten content in the Ni-W coating causes linear increase of *CPD*_av_. What is more, the corrosion rate (represented by *j*_corr_) of Ni-W coatings linearly decreases with increasing *CPD*_av_. Thus, *CPD*_av_ value allows estimating the corrosion rate of Ni-W coatings after heat treatment in air.

Figure 8 shows surface topography maps of the heat-treated Ni-W coatings obtained at deposition current density 125 mA∙cm^−2^ (a) and 200 mA∙cm^−2^ (b). Maps allow determining parameters describing quantitatively the surface roughness i.e. root mean square roughness (*S*_q_), maximum peak height (*S*_p_) and maximum pit depth (*S*_v_). It was found that for the C125/1173 coating *S*_q_ = 0.8 μm, *S*_p_ = 2.9 μm, *S*_v_ = 2.6 μm and for C200/1173 coating *S*_q_ = 9.8 μm, *S*_p_ = 20.9 μm, *S*_v_ = 21.1 μm. It can be concluded that both coatings are characterized by a uniform distribution of peaks and valleys heights around the mean. However, it should be noted that for C200/1173 coating *S*_p_ and *S*_v_ parameters are 7-8 times higher in comparison with C125/1173. This fact can be explained by that as the deposition current density increases, the small globules visible on the C125/1173 surface (see Figure 1) change into larger ones. It was also stated that the deviation of peaks and valleys heights around the mean (*S*_q_ parameter) for C200/1173 is higher. This is due to the fact that the *S*_q_ parameter is directly related to the heights of peaks and valleys on the material surface.

## 4. Conclusions

It was found that C125/1173 coating is the most resistant to corrosion in 5 wt.% NaCl solution of all the coatings tested. This is evidenced by the highest values of the corrosion potential, average contact potential difference and polarization resistance as well as the lowest value of the corrosion current density. The reason for this is the highest tungsten content in C125/1173 and the smallest surface area of this coating. Analysis of contact potential difference distribution shows also that the C125/1173 coating is characterized by the most homogeneous surface of all the coatings tested.

## Figures and Tables

**Figure 1 materials-13-01172-f001:**
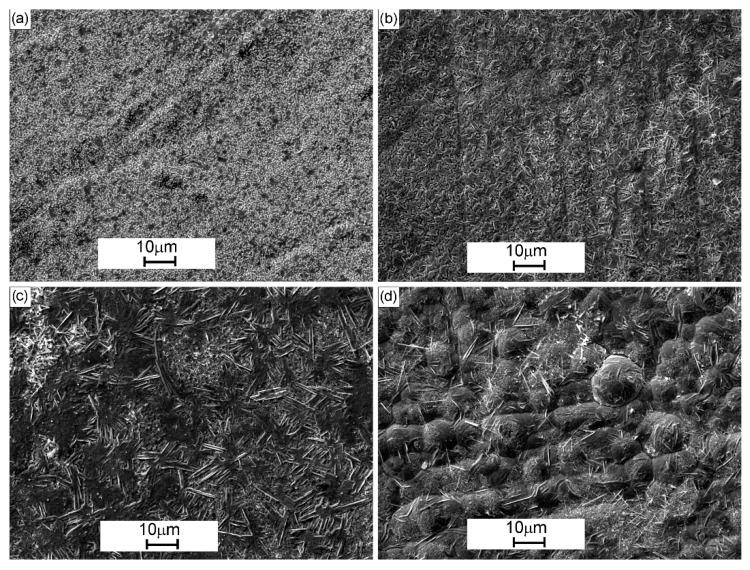
Surface morphology of Ni-W coatings after heat treatment in the air, in dependence on deposition current density: (**a**) C125/1173, (**b**) C150/1173, (**c**) C175/1173 and (**d**) C200/1173.

**Figure 2 materials-13-01172-f002:**
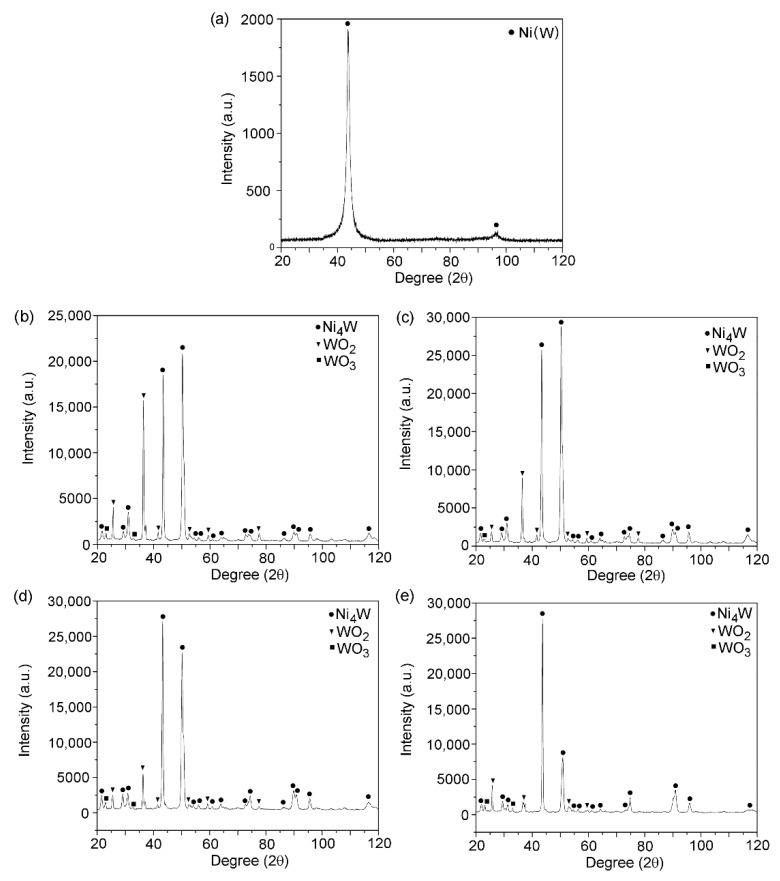
X-ray diffraction patterns for the as-deposited (**a**) C175/- and after heat treatment (**b**) C125/1173, (**c**) C150/1173, (**d**) C175/1173, (**e**) C200/1173 Ni-W coatings.

**Figure 3 materials-13-01172-f003:**
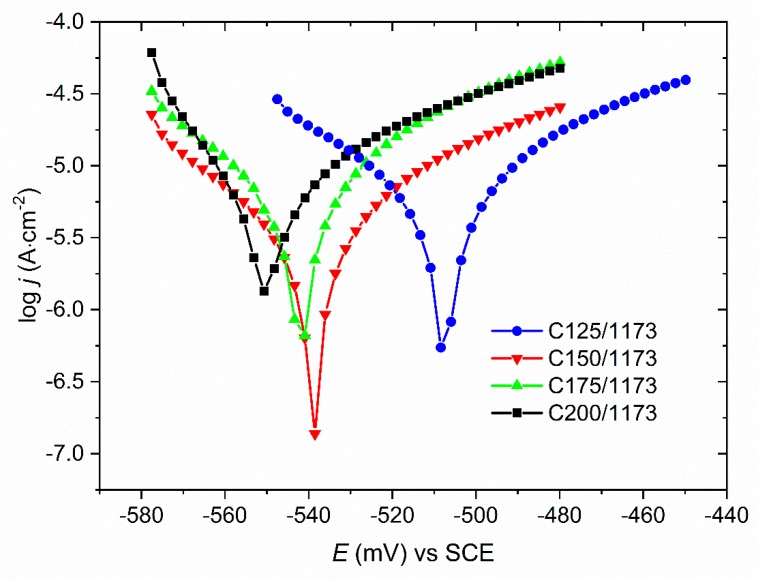
Potentiodynamic curves registered in 5 wt.% NaCl solution for the Ni-W coatings after heat treatment in the air, in dependence on deposition current density.

**Figure 4 materials-13-01172-f004:**
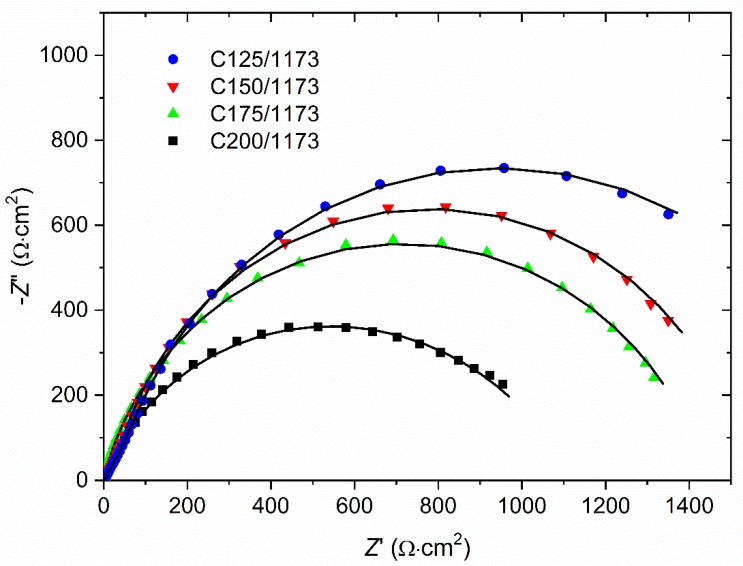
Nyquist plots registered in 5 wt.% NaCl solution for the Ni-W coatings after heat treatment in the air, in dependence on deposition current density; symbols–experimental points, solid lines–approximations using the one-CPE model.

**Figure 5 materials-13-01172-f005:**
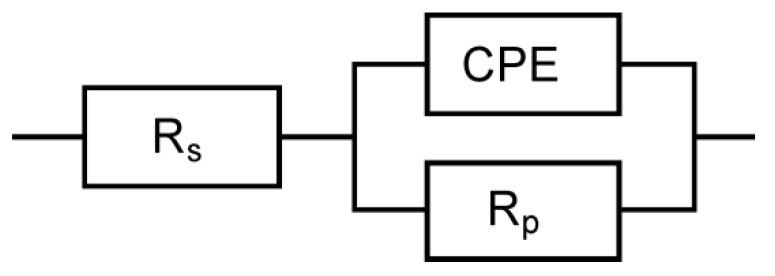
Equivalent circuit scheme, *R*_s_–solution resistance, *CPE*–constant phase element, *R*_p_–polarization resistance.

**Figure 6 materials-13-01172-f006:**
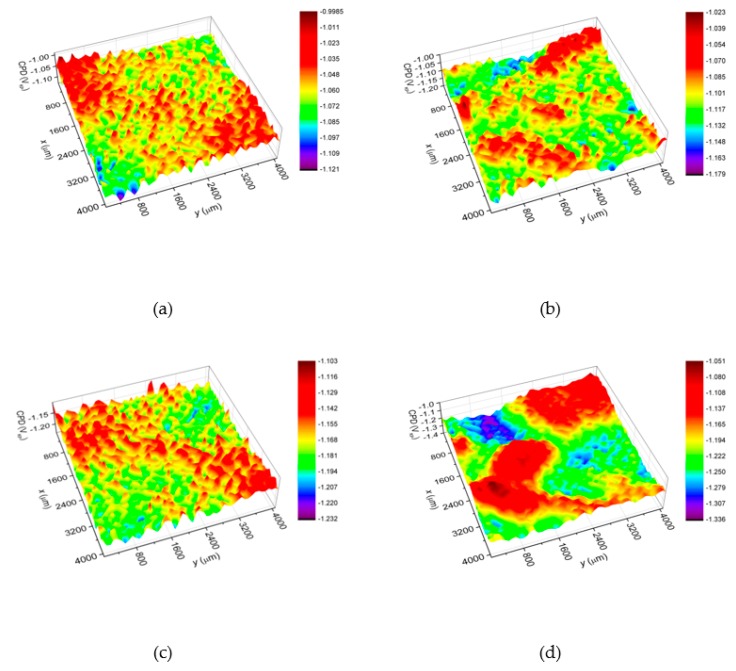
*CPD* maps determined for the Ni-W coatings after heat treatment in the air: (**a**) C125/1173, (**b**) C150/1173, (**c**) C175/1173 and (**d**) C200/1173.

**Figure 7 materials-13-01172-f007:**
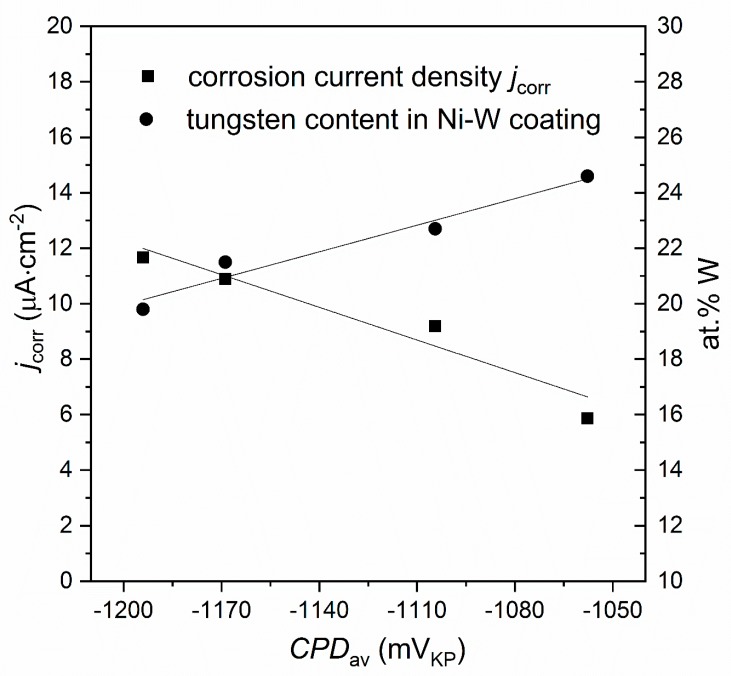
Tungsten content (at.%) and corrosion current density (*j*_corr_) versus average contact potential difference (*CPD*_av_) determined for the Ni-W coatings; mV_KP_ is the voltage measured in relation to the Kelvin probe.

**Figure 8 materials-13-01172-f008:**
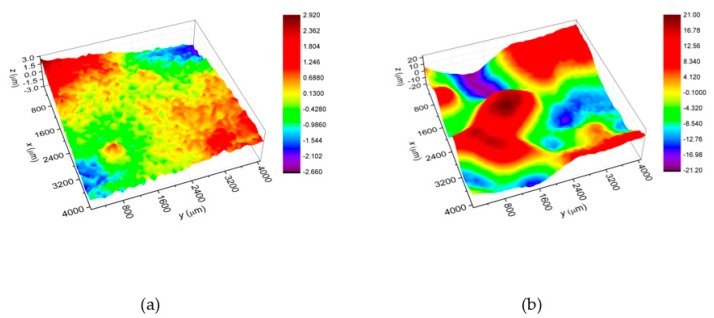
Topography maps determined for the heat-treated Ni-W coatings: (**a**) C125/1173 and (**b**) C200/1173.

**Table 1 materials-13-01172-t001:** Chemical composition of the as-deposited Ni-W alloy coatings determined by energy dispersive spectroscopy, in dependence on deposition current density.

Type of As-Deposited Coatings	At.% Ni	At.% W
Ni-W (*j*_dep_ = 125 mA∙cm^−2^)	75.4 ± 0.4%	24.6 ± 0.4%
Ni-W (*j*_dep_ = 150 mA∙cm^−2^)	77.3 ± 0.2%	22.7 ± 0.2%
Ni-W (*j*_dep_ = 175 mA∙cm^−2^)	78.5 ± 0.7%	21.5 ± 0.7%
Ni-W (*j*_dep_ = 200 mA∙cm^−2^)	80.2 ± 0.1%	19.8 ± 0.1%

**Table 2 materials-13-01172-t002:** Corrosion potential *E*_corr_ and corrosion current density *j*_corr_ determined for Ni-W coatings after heat treatment in the air, in dependence on deposition current density.

Ni-W Coating	*E*_corr_ (mV)	*j*_corr_ (μA∙cm^−2^)
C125/1173	−508	5.9
C150/1173	−538	9.2
C175/1173	−539	10.9
C200/1173	−550	11.7

**Table 3 materials-13-01172-t003:** EIS parameters determined for Ni-W coatings after heat treatment in the air, in dependence on deposition current density.

Ni-W Coating	*R*_p_ (kΩ∙cm^2^)	*T*	*ϕ*	*R*_s_ (Ω∙cm^2^)	*R* _f_
C125/1173	1.845	0.000146	0.87	1.19	2.00
C150/1173	1.535	0.000274	0.89	1.79	5.33
C175/1173	1.451	0.000328	0.89	1.51	6.39
C200/1173	1.122	0.000488	0.86	1.27	7.33

*R*_p_ is the polarization resistance, *T* is the capacitive parameter, *ϕ* is the parameter related to the rotation of the complex plane plot, *R*_s_ is the solution resistance, *R*_f_ is the factor of electrochemically active surface area.

**Table 4 materials-13-01172-t004:** Statistical parameters obtained using *CPD* maps of the heat-treated Ni-W coatings.

Ni-W Coating	C125/1173	C150/1173	C175/1173	C200/1173
*CPD*_av_ (mV_KP_)	−1058	−1104	−1169	−1194
*CPD*_q_ (mV_KP_)	16	22	17	55

*CPD*_av_—average value, *CPD*_q_—root mean square, mV_KP_ is the voltage measured in relation to the Kelvin probe.

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
