# Peer review of "Corrosion Resistance of Heat-Treated Ni-W Alloy Coatings"

_materials, 2020, doi:10.3390/ma13051172_

Round 1

Reviewer 1 Report

the manuscript might be accepted in a present form.

Author Response

Response to Reviewer 1 Comments

Dear Sir or Madam,

The Authors thank you very much for the review.

Reviewer 2 Report

This manuscript is written fairly well, comprising the experimental material with interpretation.

The authors are advised to consider the following observation:

- The novelty of this work must be better highlighted both in introduction and in experimental sections of the manuscript. This aspect is very important since they have cited more than 20 different papers (including some of their works) which refer to different aspects regarding the use of Ni-W alloy coatings. What does this study add to the previous ones? This aspect should be found in the introduction.

- In the Materials and Methods section of the manuscript, in Table 1 are presented some data about the chemical composition of the as-deposited Ni-W alloy coatings, in dependence on deposition current density. The authors are asked to present or explain here in which way they determined the composition in Ni and W of the deposited coatings. If this information was already published elsewhere, the corresponding citation should be mentioned here.

- Please provide figures with higher resolution to replace Figure 3, Figure 4, Figure 6 and Figure 8.

- Authors are kindly asked to review the entire manuscript because there are some spellings and grammar errors.

By summing up, I recommend this study for publication in Materials after minor revision.

Good luck for the authors!

Author Response

Response to Reviewer 2 Comments

Reviewer:
This manuscript is written fairly well, comprising the experimental material with interpretation.

Dear Sir or Madam,
We are most grateful for your opinion about the suitability of the manuscript attached. Below you will find reply to all your comments and suggestions. All the changes made have been highlighted in the attached revised version of the manuscript. 

Reviewer:
The novelty of this work must be better highlighted both in introduction and in experimental sections of the manuscript. This aspect is very important since they have cited more than 20 different papers (including some of their works) which refer to different aspects regarding the use of Ni-W alloy coatings. What does this study add to the previous ones? This aspect should be found in the introduction.

The novelty of this work was highlighted in the Introduction section of the manuscript.

Reviewer:
In the Materials and Methods section of the manuscript, in Table 1 are presented some data about the chemical composition of the as-deposited Ni-W alloy coatings, in dependence on deposition current density. The authors are asked to present or explain here in which way they determined the composition in Ni and W of the deposited coatings. If this information was already published elsewhere, the corresponding citation should be mentioned here.

In the Materials and Methods section of the manuscript the Authors pointed out how the chemical composition of the coatings was determined.

Reviewer:
Please provide figures with higher resolution to replace Figure 3, Figure 4, Figure 6 and Figure 8.

The figures 3, 4, 6 and 8 have been replaced by new figures with higher resolution.

Reviewer:
Authors are kindly asked to review the entire manuscript because there are some spellings and grammar errors.

In addition, the Authors improved the work in terms of language.

Reviewer 3 Report

In the present manuscript authors reported the structural characterization of Ni-W alloy which was prepared by electrodeposition followed by heat treatment under air. The results show that different current density that are applied to synthesize the materials influence their morphology and resistance towards corrosion in 5 wt% NaCl solution. Although the results shown here is interesting, the manuscript cannot be considered for the publication in the present form. A significant improvement is necessary for further consideration.

Please find below the comments and suggestions that needs to be addressed in detail.  

  1. Authors shown that at% of W is changed across the different materials, which was further correlated with their resistance to electrochemical corrosion. But the values reported here are not very different to each other. Authors should provide standard deviation of their measurement.
  2. Please report the current density either mA/cm2 or A/cm2 throughout the manuscript.
  3. Line 91, authors mentioned that structure of C125/1173 shows small separately located globules whereas the C200/1173 possess the larger one. However, C125/1173 show smaller roughness factor. This is counterintuitive and authors should clarify this.
  4. Why authors choose 1173 K as their temperature for the heat treatment?
  5. Authors should also add some discussion regarding why different current density led to different morphology of the materials.
  6. It is apparent from the XRD pattern that, there are two major peaks corresponding to Ni4W located at around 2θ 43 and 50. However, the intensity ratio of these two peaks are different at different materials. Authors should provide an explanation for this.
  7. XRD data suggested that WO2 is the major tungsten oxide component in these materials. However, why the intensity changes at different materials?
  8. Why higher tungsten content provides better corrosion resistance?

Author Response

Response to Reviewer 3 Comments

Reviewer:
In the present manuscript authors reported the structural characterization of Ni-W alloy which was prepared by electrodeposition followed by heat treatment under air. The results show that different current density that are applied to synthesize the materials influence their morphology and resistance towards corrosion in 5 wt% NaCl solution. Although the results shown here is interesting, the manuscript cannot be considered for the publication in the present form. A significant improvement is necessary for further consideration.

Dear Sir or Madam,
We are most grateful for your opinion about the suitability of the manuscript attached. Thank you very much for all your comments and suggestions, which we find very relevant and useful, also for our future scientific work. Below you will find a point-by-point reply to all your comments and suggestions. All the changes made have been highlighted in the attached revised version of the manuscript. 

Reviewer:
Authors shown that at% of W is changed across the different materials, which was further correlated with their resistance to electrochemical corrosion. But the values reported here are not very different to each other. Authors should provide standard deviation of their measurement.

Standard deviation of measurement is given in Table 1.

Reviewer:
Please report the current density either mA/cm2 or A/cm2 throughout the manuscript.

Current density is given in mA·cm-2 throughout the manuscript.

Reviewer:
Line 91, authors mentioned that structure of C125/1173 shows small separately located globules whereas the C200/1173 possess the larger one. However, C125/1173 show smaller roughness factor. This is counterintuitive and authors should clarify this.

The surface properties of an electrode immersed in a particular solution can change drastically after it has dried. Therefore, analysis of topographic images alone is not sufficient to characterize the nature of the electrode surface. In corrosion tests, we used in-situ electrochemical technique (EIS) to estimate an electrochemically active surface area, which we called the roughness factor. In order to avoid misunderstanding the phrase “roughness factor” has been replaced by “factor of electrochemically active surface area”.

Reviewer:
Why authors choose 1173 K as their temperature for the heat treatment?

The Authors chose 1173 K (900 oC) as temperature for the heat treatment, based on the chemical composition of the tested coatings and the phase diagram of the Ni-W system (Figure 1). The 1173 K temperature used was aimed to over reacting of the coating components and in consequence in order to obtaining a Ni4W intermetallic phase.

Figure 1. Phase diagram of the Ni-W system.

Reviewer:
Authors should also add some discussion regarding why different current density led to different morphology of the materials.

Coatings obtained at low current density values ​​have a poorly developed surface. It can be explained by that low current densities favor the slow discharge of ions at electrodes, and therefore the growth rate of the resulting grains exceeds the speed of forming of new ones. As the current density increases, the rate of formation of new grains also increases what result in more developed surface. The increase in the density of the deposition current causes intense hydrogen evolution, which in turn can cause the formation of porous coatings.

Reviewer:
It is apparent from the XRD pattern that, there are two major peaks corresponding to Ni4W located at around 2θ 43 and 50. However, the intensity ratio of these two peaks are different at different materials. Authors should provide an explanation for this.

The intensity ratio of two major reflexes corresponding to Ni4W intermetallic phase located at around 2θ = 43° and 50° are different at different materials, what may by result from coatings texture. The electrocrystallization texture is an axial texture with an axis perpendicular to the direction of electric current flow. The texture direction depends on the electrochemical conditions of the deposition process.

Reviewer:
XRD data suggested that WO2 is the major tungsten oxide component in these materials. However, why the intensity changes at different materials?

Indeed, intensity of WO2 reflexes is different for different coatings. This is due to the different tungsten content in the tested coatings. The presented X-ray patterns show that the highest intensity of the WO2 reflex is for the C125/1173 coating. This coating is also characterized by the highest tungsten content.

Reviewer:
Why higher tungsten content provides better corrosion resistance?

This is probably due to the higher tungsten contents the higher passivation degree of coating surface, see answer above - 7.

Reviewer 4 Report

The manuscript entitled "Corrosion Resistance of Heat-Treated Ni-W Alloy Coatings" describes the investigations concerning an assessment of corrosion resistance of heat-treated Ni-W alloy coatings. Resistance to electrochemical corrosion was tested using potentiodynamic polarization technique and electrochemical impedance spectroscopy. Characteristics of the Ni-W coatings after heat treatment were carried out using scanning electron microscopy, scanning Kelvin probe technique and X-ray diffraction. the manuscript is well written and characterized. Therefore, I do recommend current manuscript to be published in Masterials with just one recommendation.

The authors said that the reasons for the improvement of protective properties of the heat treated Ni-W coating are the highest tungsten content (c.a. 25 at.%) as well as the smallest and the most homogeneous electrochemically active surface area. But, in my opinion, authors should expand the discussion on why the coating has the highest tungsten content in C125/1173 and the smallest surface area either.

Author Response

Response to Reviewer 4 Comments

Reviewer:
The manuscript entitled "Corrosion Resistance of Heat-Treated Ni-W Alloy Coatings" describes the investigations concerning an assessment of corrosion resistance of heat-treated Ni-W alloy coatings. Resistance to electrochemical corrosion was tested using potentiodynamic polarization technique and electrochemical impedance spectroscopy. Characteristics of the Ni-W coatings after heat treatment were carried out using scanning electron microscopy, scanning Kelvin probe technique and X-ray diffraction. the manuscript is well written and characterized. Therefore, I do recommend current manuscript to be published in Materials with just one recommendation.

Dear Sir or Madam,
We are most grateful for your opinion about the suitability of the manuscript attached. According to your suggestion, we have extended the discussion on why the C125/1173 coating has the highest tungsten content and the smallest surface.

Reviewer:
The authors said that the reasons for the improvement of protective properties of the heat treated Ni-W coating are the highest tungsten content (c.a. 25 at.%) as well as the smallest and the most homogeneous electrochemically active surface area. But, in my opinion, authors should expand the discussion on why the coating has the highest tungsten content in C125/1173 and the smallest surface area either.

The changes made have been highlighted in the attached revised version of the manuscript (lines 97 – 102).

Round 2

Reviewer 3 Report

The authors provided the satisfactory answer to all my queries and I now recommend the present manuscript for publication. 

Author Response

The authors provided the satisfactory answer to all my queries and I now recommend the present manuscript for publication. 

Answer: We thank the expert reviewer for his/her comment, no further changes are required.